# Engineering an artificial catch bond using mechanical anisotropy

Zhaowei Liu [1,2,7], Haipei Liu [1,2], Andrés M. Vera [3], Byeongseon Yang [1,2,4,5], Philip Tinnefeld [3] & Michael A. Nash [1,2,4,5,6] ✉

Catch bonds are a rare class of protein-protein interactions where the bond lifetime increases under an external pulling force. Here, we report how modification of anchor geometry generates catch bonding behavior for the mechanostable Dockerin G:Cohesin E (DocG:CohE) adhesion complex found on human gut bacteria. Using AFM single-molecule force spectroscopy in combination with bioorthogonal click chemistry, we mechanically dissociate the complex using five precisely controlled anchor geometries. When tension is applied between residue #13 on CohE and the N-terminus of DocG, the complex behaves as a two-state catch bond, while in all other tested pulling geometries, including the native configuration, it behaves as a slip bond. We use a kinetic Monte Carlo model with experimentally derived parameters to simulate rupture force and lifetime distributions, achieving strong agreement with experiments. Single-molecule FRET measurements further demonstrate that the complex does not exhibit dual binding mode behavior at equilibrium but unbinds along multiple pathways under force. Together, these results show how mechanical anisotropy and anchor point selection can be used to engineer artificial catch bonds.

While conventional slip bonds weaken under the influence of mechanical forces, catch bonds summon conformational changes that strengthen bonding and enhance the bond lifetime[1,2]. This unique behavior allows cells to adhere tightly under high shear conditions but dissociate and move freely when flow subsides. In recent years the physiological relevance of catch bond mechanics has been widely appreciated, with many prominent examples of protein receptor-ligand catch bonds emerging[3-14]. Atomic force microscopy operated in single-molecule force spectroscopy mode (AFM-SMFS) is a well-established method for quantifying catch bond behavior[4,5,11]. AFM-SMFS performed in force clamp mode can directly measure the lifetime of protein receptor-ligand complexes, and the measured lifetime of catch bonds typically increases with increasing clamping force in the catch activation regime. However, force clamp methodology can be challenging to implement because it requires high thermal and vibrational stability in the experimental setup, as well as a fast and well-tuned feedback system[4,7,15]. Alternatively, AFM-SMFS in force ramp or constant speed mode is more straightforward to implement and can be equivalently used to quantify catch bond behavior[16]. In this measurement mode, catch bonds characteristically exhibit bimodal rupture force distributions where the prevalence of the high force population increases at higher loading rates[11,17]. This discrete pathway switching is distinct from the conventional log-linear loading rate dependency of slip bonds as described by the Bell–Evans model[18,19] and manifests as a discontinuous jump in the most probable rupture force (i.e., switch to a high-force pathway) that takes place over a range of activation forces.

The ability to strengthen under external force makes catch bonds potentially applicable for targeted cell and nanoparticle therapies[20].

---

[1]Institute of Physical Chemistry, Department of Chemistry, University of Basel, 4058 Basel, Switzerland. [2]Department of Biosystems Science and Engineering, ETH Zurich, 4058 Basel, Switzerland. [3]Faculty of Chemistry and Center for NanoScience, Ludwig-Maximilians-Universität München, Munich, Germany. [4]Botnar Research Centre for Child Health, 4051 Basel, Switzerland. [5]National Center for Competence in Research (NCCR) Molecular Systems Engineering, 4058 Basel, Switzerland. [6]Swiss Nanoscience Institute, 4056 Basel, Switzerland. [7]Present address: Department of Bionanoscience, Delft University of Technology, 2629HZ Delft, the Netherlands. ✉e-mail: michael.nash@bsse.ethz.ch

However, engineering of artificial catch bonds in an analogous manner to current methods for synthetic antibody selection is currently not possible because molecular structural mechanisms enabling catch bonding are poorly understood, no generalizable scaffolds are known, and design heuristics are unavailable.

One concept that is known to play a critical role in molecular deformation under force is that of mechanical anisotropy. This concept refers to the fact that in order for mechanical force to act on proteins, molecules must be biochemically tethered to a force transducing element such as a cell surface or in the case of in vitro experiments to a micro-to-nanoscale force transducer (e.g., AFM cantilever). Differences in anchor geometries of biomolecules strongly impact their unfolding and unbinding stability under force, and these effects have been widely shown in the shear *vs.* unzipping behavior of nucleic acids[21–24], for example, along with numerous studies on mechanically anisotropic unfolding of single protein domains[25–27] and protein receptor-ligand complexes[28–31]. The mechanostabilities of native catch bonds are also known to be regulated by anchor directions[32,33]. Despite widespread recognition that directional pulling has a deterministic influence on mechanical responses in biomolecules, this approach has not been pursued for engineering synthetic non-native catch bonds.

Here, we used AFM-SMFS combined with bioorthogonal protein engineering to study the anchor geometry-dependence of the mechanical response of the *Ruminococcus champanellensis* (*Rc.*) Doc-G:CohE complex. Docs and Cohs comprise a family of highly mechanostable bacterial adhesion complexes[34], which direct the self-assembly of large extracellular cellulose-degrading protein networks called cellulosomes. We combined AFM-SMFS with click chemistry[31] to dissociate the *Rc.* DocG:CohE complex[35] in five different well-controlled pulling configurations and found that the complex is able to generate catch bond behavior under a specific non-natural anchor geometry, whereas the other anchor geometries, including the native anchor geometry, gave rise to slip bonding. The distinct slip and catch bond behaviors found for different anchor geometries were further validated by kinetic Monte Carlo simulations. Furthermore, we used single-molecule FRET to show that the two distinct unbinding pathways originated from a single binding conformation, excluding the possibility of dual binding modes. Our mechanical anisotropy-based protein engineering approach can significantly alter unbinding pathways under external forces and introduce new mechanical properties such as catch bonding, while requiring only a single amber codon substitution in primary sequence to introduce an anchor point.

## Results

### The *Rc.* Doc:Coh complex dissociates along two pathways in the native anchor geometry

There is no available structural information for the *Rc.* DocG:CohE complex, therefore we used homologous modeling approaches to build structural models for the complex. The structural model of DocG was built using the SWISS-MODEL server (template: PDB 4WKZ)[36–41], and the DocG structural model was aligned to a previously reported *Rc.* XMod-DocB:CohE structural model[11] using PyMOL to build a model for the DocG:CohE complex, as shown in Fig. 1a. DocG has 29% sequence identity with the modeling template and 35% sequence identity with the alignment template.

The native anchor geometry of the *Rc.* DocG:CohE complex in vivo is at the N-terminus of DocG and C-terminus of CohE[35,41]. To mimic the native anchor geometry on AFM, DocG and CohE were cloned into bacterial expression vectors as fusion proteins (ybbr-ELP-FLN-DocG and CohE-FLN-ELP-ybbr) and expressed in *E. coli*. The intrinsically disordered elastin-like peptide (ELP) was used as an elastic linker to separate the proteins from the surfaces[42]. The filamin (FLN) domain served as a fingerprint to help identify single-molecule interactions[43,44]. The ybbr tag was used to covalently immobilize DocG and CohE on the coenzyme A (CoA)-coated glass surface and AFM tip, respectively[45]

(Fig. 1b). The AFM cantilever approached the surface and dwelled there for 200 ms to form a DocG:CohE complex between the tip and the glass surface. The cantilever was then retracted with a constant speed, ranging between 100 nm s$^{-1}$ and 3200 nm s$^{-1}$, to apply pulling force to the DocG:CohE complex from the N-terminus of DocG and the C-terminus of CohE, precisely mimicking the native pulling geometry. A force-extension curve was recorded in each approach-retraction cycle. Over 10,000 curves were typically recorded in an overnight measurement (~12 h). Raw curves were transformed into contour length space using the freely rotating chain (FRC) elasticity model (Eq. (1))[46]. Each ELP linker in the DocG and CohE constructs was 170 amino acids long, and the total contour length of two ELP linkers was 170 × 2 × 0.365 nm/amino acid = 124 nm. The force curves were filtered based on the contour length at the time of complex rupture. Only force curves showing fully stretched ELP linkers (i.e. total contour length at final rupture >124 nm) were taken for further analysis to exclude non-specific interactions between the tip and the surface. A majority of the filtered force curves showed unfolding events corresponding to two FLN fingerprint domains, as shown in Fig. 1c (top panel). Each FLN unfolded in two steps and added ~32 nm of contour length to the system upon unfolding. Unfolding of both FLNs therefore gave rise to ~64 nm of contour length increment, which clearly emerged in the combined contour length histogram of all filtered curves (Fig. 1d)[47,48]. A small fraction of the force curves (<5%) did not ramp up to sufficient force to unfold the FLN domains (Fig. 1c, middle and bottom panels).

The rupture forces of the DocG:CohE complex when pulled in the native anchor geometry (N-term. DocG, C-term. CohE) were measured at four different pulling speeds (100, 400, 800 and 3200 nm s$^{-1}$) and plotted in histograms (Fig. 1e). The histograms clearly showed bimodal distributions, suggesting two different unbinding pathways with distinct rupture forces. The rupture force histograms were fitted with two gaussian distributions to extract the most probable rupture force for each unbinding pathway, which were fitted linearly against the logarithm of loading rate to extract Bell–Evans energy landscape parameters (zero-force off rate $k_O$ and distance to the transition state $\Delta x^{\ddagger}$) as described by Eq. (2)[18,19]. The extracted energy landscape parameters are presented in Table 1 (see first row, Native anchor geometry (CohE wild-type construct)).

### Single-molecule FRET shows a single binding conformation

Several Doc:Coh complexes are known to exhibit dual-binding mode behavior, meaning they can assemble in two distinct binding conformations which differ by a 180° rotation of Doc with respect to the binding surface on Coh[11,49–53]. This unique dual binding capability is difficult to resolve with ensemble methods since both modes have similar affinity, therefore single-molecule methods are well suited. To determine if the two pathways observed on AFM-SMFS were the result of two binding modes, we used single-molecule FRET (smFRET), which provides relative distance information[54–56] and has been used to resolve dual-binding modes in other Doc:Coh systems[11,53]. We first built structural models of our DocG:CohE complex in two putative conformations labeled A (shown in Figs. 1a and 2a) and B (shown in Fig. 2a). These two models were constructed using the two *Rc.* XMod-Doc-B:CohE binding conformations, respectively, as alignment templates[11]. Based on these models, positions were chosen for dye conjugation for smFRET experiments which would maximize the difference in dye-to-dye distance between the two binding modes.

The *Rc.* CohE was labeled with the FRET acceptor, Alexa Fluor 647, at the C-terminus (residue E154) and the DocG was labeled with the FRET donor, Cy3b, at the helix 2 (residue S54) or the C-terminus (residue A77) (see Fig. 2a, b). The residue at the labeling site was mutated to cysteine and covalently linked to a maleimide-conjugated fluorophore. The labeled CohE and DocG were mixed at picomolar concentration and the proximity ratios of individual DocG:CohE

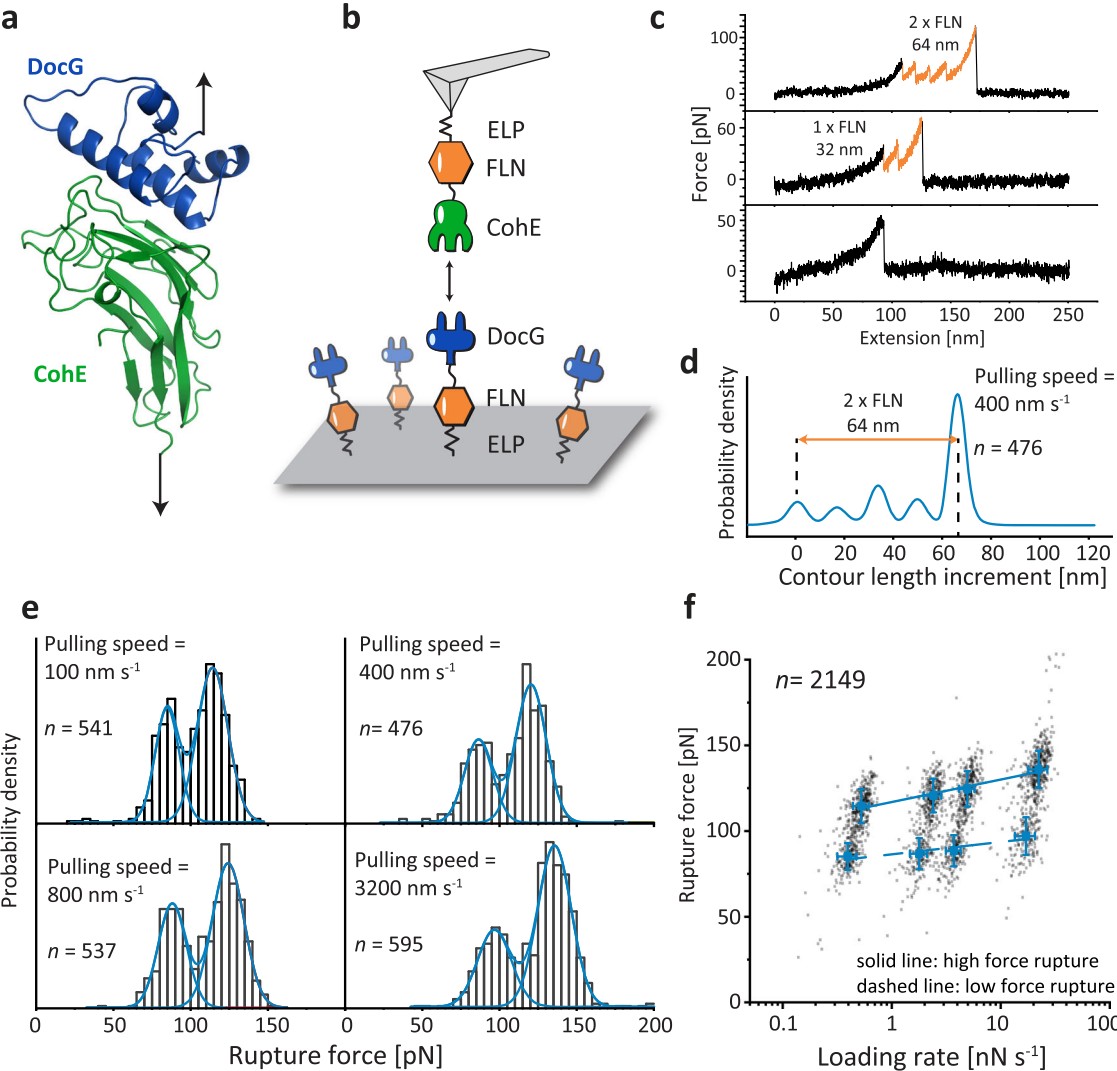

**Fig. 1 | Structural model and AFM measurements of the DocG:CohE complex in the native pulling geometry. a** Structural model of *Rc.* DocG:CohE complex. The DocG model was built with the SWISS-MODEL server and aligned to the structure of *Rc.* XMod-DocB:CohE to build a structural model of the DocG:CohE complex. **b** AFM-SMFS measurement setup for the native anchor geometry. The CohE-FLN-ELP-ybbr and ybbr-ELP-FLN-DocG constructs were site-specifically immobilized using ybbr tags. An AFM tip approached the glass surface and was retracted to dissociate the DocG:CohE complex in the native anchor geometry. **c** Example force-extension curves. Most of the curves ramped up to sufficient force to unfold both FLN domains, giving rise to a 64 nm contour length increment (top panel). Less than 5% of the curves contained zero or one FLN unfolding events due to Doc:Coh rupture prior to the unfolding of one or two FLNs. **d** Combined contour length

histogram of all force-extension curves ($n = 476$) measured at a pulling speed of 400 nm s$^{-1}$. The histogram shows a contour length increment of 64 nm between the first FLN unfolding step and the final rupture of the complex. **e** Rupture force histograms of DocG:CohE measured in the native anchor geometry at four pulling speeds. Each histogram was fitted with two Gaussian peaks to calculate the most probable rupture force of each population. **f** Force-loading rate plot of $n = 2149$ independently measured single protein complexes. The most probable rupture forces of each unbinding pathway were plotted and linearly fit (high force: solid line, low force: dashed line) against the logarithm of the loading rate to extract energy landscape parameters ($k_0$ and $\Delta x^{\ddagger}$). Error bars show the standard deviation of loading rates and forces. The centers of the error bars represent the most probable rupture force and the mean loading rate of each pulling speed.

complexes were measured using a confocal microscope and plotted in histograms, as shown in Fig. 2c. The proximity ratios measured with FRET donor labeled at either S54 or A77 of DocG showed unimodal distributions, demonstrating that the *Rc.* DocG:CohE complex populates only one binding conformation.

Based on the smFRET data, we sought to determine which putative binding conformation (A or B) was populated by the *Rc.* DocG:CohE complex. The distances between the FRET acceptor site on CohE and the donor labeling sites on DocG are different in the two possible conformations. Therefore, we used photon distribution analysis (PDA) to fit the distance between FRET donor and acceptor (see Supplementary Fig. S1)[57,58]. The results showed that the distance between residues CohE E154 and DocG S54 was longer than the distance between CohE E154 and DocG A77. This was consistent with the

putative binding conformation A shown in Figs. 1a and 2a, and not with the putative binding conformation B shown in Fig. 2b. Based on this analysis, we concluded that the *Rc.* DocG:CohE complex populates only binding mode A, and cannot assemble in binding mode B. The two pathways observed for the native anchoring geometry in the AFM-SMFS were therefore attributed to competing unbinding pathways from a single bound conformation.

## A non-native anchor geometry gives rise to catch bond behavior

Given the high-force and low-force unbinding pathways that we observed in *Rc.* DocG:CohE, we hypothesized that this complex would be a valid candidate for catch bond engineering based on anchor point selection. We tested various combinations of anchor points in DocG and CohE that could result in pathway switching from the low-force to

**Table 1 | Energy landscape parameters of Doc:Coh complex at different pulling geometries**

| Anchor geometry code | DocG anchor geometry | CohE anchor geometry | Pathway | $\log(k_O)$ | $\Delta x^{\ddagger}$ [nm] |
|---|---|---|---|---|---|
| Native | N-terminus | C-terminus (wild-type construct) | High force | −7 ± 1 | 0.73 ± 0.09 |
| | | | Low force | −9 ± 3 | 1.3 ± 0.4 |
| Native | N-terminus | C-terminus (CohE F13AzF-Fgβ construct, as in Supplementary Fig. S5a) | High force | −4.7 ± 0.4 | 0.54 ± 0.03 |
| | | | Low force | −5.5 ± 0.5 | 0.83 ± 0.06 |
| NN-A | C-terminus | C-terminus | High force | −7 ± 2 | 1.1 ± 0.2 |
| | | | Low force | −31 ± 7 | 10 ± 2 |
| NN-B | N-terminus | N-terminus | N/A | −4.2 ± 0.3 | 0.65 ± 0.04 |
| NN-C | N-terminus | F13 | High force | −2.5 ± 0.6 | 0.37 ± 0.06 |
| | | | Low force | −0.8 ± 0.2 | 0.51 ± 0.04 |
| NN-D | N-terminus | K87 | High force | −4.8 ± 0.6 | 0.58 ± 0.06 |
| | | | Low force | −0.7 ± 0.3 | 0.41 ± 0.05 |

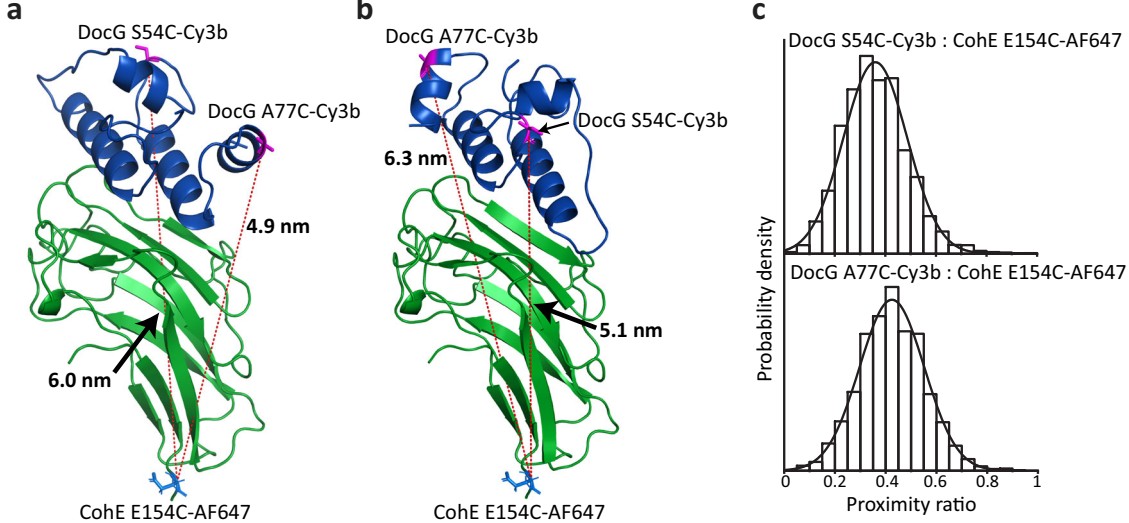

**Fig. 2 | Single-molecule FRET demonstrates a single binding mode. a, b** Two putative binding conformations of DocG:CohE complex. The FRET donor (Cy3b) was conjugated to one of the two labeling sites of DocG (S54 or A77). The FRET acceptor (Alexa Fluor 647) was conjugated to the residue E154 of Coh. **c** smFRET measurements with the donor labeled at residue S54 (upper panel) or A77 (lower panel) of Doc showed a unimodal distribution of proximity ratio, indicating that the complex populates one binding conformation at equilibrium.

high-force at increased loading rate in AFM-SMFS, precisely mimicking a catch bond. To pursue this, we combined AFM-SMFS with click chemistry to dissociate the DocG:CohE complex by pulling at four non-native anchor points: the C-terminus of DocG (Non-Native anchor geometry A, NN-A), the N-terminus of CohE (NN-B), and internal residues F13 (NN-C) and K87 (NN-D) of CohE (see Fig. 3a and Table 1). The anchor points were selected to cover both the residues at the edge and in the middle of the molecules, where the complex was ruptured by shearing and peeling, respectively. In these measurements, either the DocG or the CohE was pulled from the selected non-native anchor point, and the binding partner was immobilized through the native anchor point (N-terminus of DocG, C-terminus of CohE).

For the anchor points at the C-terminus of DocG and N-terminus of CohE, DocG-FLN-ELP-ybbr and ybbr-ELP-FLN-CohE constructs were cloned into bacterial expression vectors and expressed using *E. coli*. The AFM setup using tethered receptor-ligand constructs, similar to the native anchor geometry measurement (Fig. 1b), was used for these anchor geometries. The AFM cantilever was approached to the surface and dwelled for 200 ms to form a Doc:Coh complex and retracted at a constant speed (ranging between 100–3200 nm s⁻¹) to dissociate the complex in the chosen anchor geometry and record force-extension curves.

To apply tension to the complex through internal residues of CohE (F13 and K87), we combined click chemistry and a freely diffusing receptor-ligand system (Fig. 3b, c)[31]. The desired anchor residue on CohE (F13 or K87) was replaced by the noncanonical amino acid azido-phenylalanine using amber codon suppression[59]. The azide at the selected anchor point was covalently conjugated with a synthetic Fgβ-StrepTag-dibenzocyclooctyne (DBCO) peptide using the click reaction between the azide and DBCO groups. The successful conjugation was confirmed by SDS-PAGE (Supplementary Fig. S2). Fgβ is the N-terminus peptide of the human fibrinogen β chain, which binds SdrG, a bacterial adhesin from *Staphylococcus epidermidis*. The CohE conjugated with Fgβ was purified using size-exclusion and StrepTrap columns to remove the excess peptide and unreacted CohE. Microscale thermophoresis (MST) measurements showed that the AzF mutation and Fgβ conjugation did not significantly affect the equilibrium binding affinity of DocG:CohE complexes (see Supplementary Table S1). For the AFM setup, an SdrG-FLN-ELP-ybbr polyprotein was immobilized on the AFM cantilever and ybbr-ELP-FLN-DocG was immobilized on the glass surface. The biorthogonal-conjugated CohE-Fgβ were added to the AFM measurement buffer to a final concentration of ~100 nM. As the AFM cantilever approached the surface and dwelled for 200 ms, a ternary complex was formed consisting of SdrG, Fgβ-StrepTag-CohE and DocG. The cantilever was subsequently retracted at a constant speed

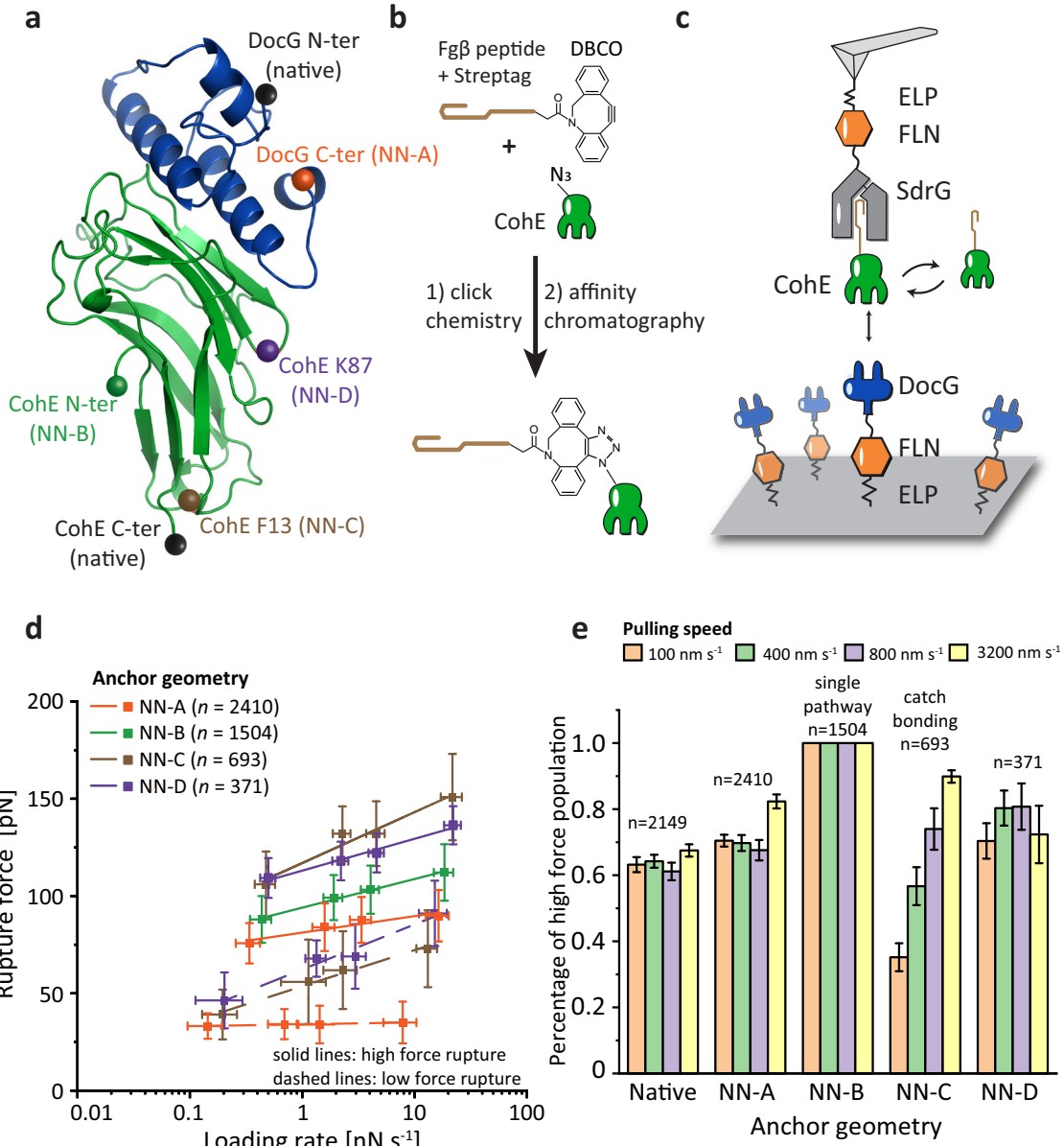

**Fig. 3 | Catch bond behavior in a non-native pulling geometry. a** Non-native anchor points selected on the DocG (C-terminus) and CohE (N-terminus, residue F13 and residue K87). **b** For internal anchor points on the CohE, the selected anchor residue was replaced with azido-phenylalanine using amber codon suppression. Fgβ was covalently conjugated with CohE at the anchor point via the click reaction between the azide and DBCO groups. **c** AFM measurement setup for internal anchor points. CohE conjugated with Fgβ was added to the AFM measurement buffer and bound with the SdrG immobilized on the AFM tip. DocG was immobilized on the glass surface at the native anchor point (N-terminus). The AFM cantilever was approached to the glass surface to form a ternary SdrG:Coh:Doc complex, and subsequently retracted at constant speed to dissociate the DocG:CohE complex in the selected anchor geometry. The rupture force of SdrG:Fgβ is -20 fold higher than DocG:CohE and therefore the final rupture events on force-extension curves report DocG:CohE rupture. **d** Rupture force *vs.* loading rate plots of non-native anchor geometries. The rupture forces exhibited unimodal NN-B) or bimodal (all other pulling geometries) distributions and the most probable rupture force of

each population was fitted against loading rate using Eq. (2) to extract $k_O$ and $\Delta x^\ddagger$ (solid line: high force population, dashed line: low force population). The error bars represent standard deviation of rupture force and loading rate at a given pulling speed. The centers of the error bars represent the mean rupture force and mean loading rate at each pulling speed. The *n* numbers represent the number of single protein complexes measured at each anchor geometry. The raw force-loading rate data are listed in the source data file. **e** Percentage of high rupture force population measured at different anchor points and different pulling speeds. The *n* numbers represent the number of single protein complexes measured at each anchor geometry. The error bars represent the standard deviation of the high force population percentage calculated based on Gaussian error propagation. The centers of the error bars represent the percentage of high force population calculated based on the area under the peaks of rupture force histograms (see Methods – AFM data analysis). The anchor geometry NN-B gave rise to a unimodal rupture force distribution and the percentage of high force population at this anchor geometry is considered to be always one.

(ranging between 100–3200 nm s⁻¹) to apply tension to the DocG:CohE complex from the N-terminus of DocG and the selected internal residue of CohE (F13 or K87) until the complex ruptured. The rupture force of the Fgβ:SdrG complex is around 2 nN[3], which is -20 fold higher than the DocG:CohE rupture force. Therefore the final rupture event

recorded on the force-extension curves always came from the DocG:CohE complex, as the system could not ramp up to sufficiently high forces to rupture the Fgβ:SdrG complex.

The collected force-extension curves were filtered using the same criterion as the native pulling geometry measurement, i.e. all the

selected force curves contained the stretching of a single pair of ELP linkers. The rupture forces of DocG:CohE complexes at different non-native anchor geometries were plotted against loading rate (Fig. 3d). Similar to the native anchor geometry, all the non-native anchor geometries, except pulling from the C-terminus of CohE (anchor geometry NN-B), gave rise to bimodal distributions of rupture forces (Supplementary Fig. S3). The most probable rupture force of each population was linearly fitted against the logarithm of loading rate to extract $k_O$ and $\Delta x^\ddagger$ using Bell−Evans model (see Eq. (2) and Table 1)[18,19]. The rupture force distributions were fitted with one or two Gaussian peaks. For each anchor geometry, the area under the high force peak was divided by the sum of the areas under the high force and low force peaks to calculate the prevalence of high force unbinding pathway.

As shown in Fig. 3d, e, the anchor geometry affected both the magnitude of the rupture forces and the prevalence of each unbinding pathway. Among all the anchor geometries we tested, anchoring at residue F13 of CohE and N-terminus of DocG (anchor geometry NN-C) gave rise to catch bond behavior. In this anchor geometry, the low rupture force pathway dominated at low loading rate, and the prevalence of the high rupture force pathway increased with increasing loading rate ($p < 0.05$), with the high force pathway dominant at high loading rates, as shown in Fig. 3e, Supplementary Figs. S3, S4 and Supplementary Table S2. This behavior precisely resembles several previously reported catch bonds[1,2,11,17]. The other anchor geometries, including the native geometry, gave rise to one or two unbinding pathways, where the prevalence of each pathway was independent of the loading rate (see Supplementary Fig. S4 and Supplementary Table S2). Therefore, the complex behaved as a single or two parallel competing slip bonds in these anchor geometries.

To confirm that the catch bond behavior originated from the change of anchor geometry, rather than the introduction of azido-phenylalanine and Fgβ peptide, we replaced CohE F13 with azido-phenylalanine in the native anchor geometry construct (Coh-FLN-ELP-ybbr), conjugated the azide group with Fgβ-Streptag-DBCO peptide, and used AFM-SMFS to dissociate the DocG:CohE complex by anchoring at the native anchor geometry (see Supplementary Fig. S5a). As shown in Supplementary Fig. S5b, c, the complex unbound in two distinct pathways, but the prevalence of these pathway was independent of the loading rate (Supplementary Fig. S5d), consistent with the slip bond behavior shown in the native anchor geometry measurements using wild-type CohE (Fig. 1e). In addition, $k_O$ and $\Delta x^\ddagger$ values measured using wild-type and mutated CohE proteins were similar (Table 1). We concluded that the introduction of azido-phenylalanine mutation and conjugation of Fgβ peptide did not affect the mechanical properties of the complex, and it is the change of anchor geometry that gave rise to the catch bond behavior.

### Kinetic models and Monte Carlo simulations
We developed two distinct kinetic models to describe the behavior of the DocG:CohE complex in the catch (NN-C, anchored at CohE F13 and DocG N-terminus) and slip anchor geometries. In the slip anchor geometries, the complex dissociated in two parallel non-interchangeable pathways with different sets of unbinding energy landscape parameters, leading to distinct mechanostabilities, as shown in Fig. 4a. The catch anchor geometry can be described using a two-state two-pathway catch bond model[1], as shown in Fig. 4b. Upon DocG:CohE binding, the complex initially enters a weak binding state. When external pulling forces are applied in the catch anchor geometry, the complex either unbinds at a low rupture force, or transitions to a strong state, from which the complex can rupture at a high force. The transition rate $k_{12}$ from the weak state to the strong state increases with increasing force, i.e. the catch bond is activated by pulling forces within a certain range. The force-dependent $k_{12}$ value was obtained based on the prevalence of high and low force pathways at different pulling speeds, using a nonlinear least-square fitting approach as

previously described[60]. Based on the aforementioned kinetic models, we used Monte Carlo simulations to simulate the behavior of the complex in both the native slip anchor geometry and the catch anchor geometry. It is noted that the configurations of anchor points were not taken into account in the Monte Carlo simulations, in which the different dissociation pathways are focused. We conducted constant pulling speed simulations using the same pulling speeds as used in experiments (100, 400, 800 and 3200 nm s$^{-1}$). The simulated rupture forces were plotted against the logarithm of force loading rate (Fig. 4c, d) and fitted linearly to extract $k_O$ and $\Delta x^\ddagger$ using the Bell−Evans model (Eq. (2)). The loading rate dependency of rupture force and the calculated $k_O$ and $\Delta x^\ddagger$ values, as shown in Supplementary Fig. S6 and Supplementary Table S3, are highly consistent with the experimental values (see Table 1). In addition, the simulated rupture force histograms (Supplementary Fig. S7a) showed similar behavior compared to experimental results. We observed in the simulations that the prevalence of the high rupture force population increased with force loading rate in the catch anchor geometry, but not the native anchor geometry (see Supplementary Fig. S7b and Supplementary Table S2), consistent with the experimental results.

Furthermore, we investigated the force dependency of the DocG:CohE bond lifetime using force-clamp Monte Carlo simulations. In the native anchor geometry, the prevalence of high force and low force pathways were independent of the clamping force, and the median bond lifetime decreased exponentially with clamping force (see Fig. 4e), a characteristic of slip bonds. In contrast, in the catch anchor geometry, the prevalence of the high force pathway increased at forces below ~30 pN, and the median lifetime of the interaction increased with increasing force between ~20−30 pN, which resembles a typical catch bond behavior (see Fig. 4f). At forces larger than ~30 pN, the low force pathway was completely suppressed and the complex only dissociated along the high force pathway, indicating for forces above the catch activation regime of 20−30 pN, the interaction behaved again like a slip bond. The ability to perform kinetic multi-state Monte Carlo simulations of the catch and slip bond anchor geometries using experimentally derived parameters, and the finding that the resulting simulated data closely resemble those derived from experiments provides a validation that the multi-state model and its underlying assumptions are reasonable.

## Discussion
The anisotropic response of biomacromolecules to external forces has been widely discussed for nucleic acids, protein domains, and protein-ligand complexes. Previous studies demonstrated that the shapes and heights of unfolding and unbinding energy landscapes are dependent on the anchor geometry, giving rise to distinct unfolding and rupture forces[26,31]. However, mechanical anisotropy is a concept that has not been widely exploited for synthetic protein engineering. Here, we used an AFM-SMFS experimental method combined with click chemistry to dissociate the *Rc*. DocG:CohE complex from five different pulling directions. The complex ruptured along two parallel pathways with distinct rupture forces in most of the anchor geometries. We found the anchor points affected the mechanostabilities of the various pathways as well as the probabilities of entering each of them. Among the five anchor geometries we tested, we found that anchoring CohE at residue F13 and DocG at its N-terminus gave rise to catch bond behavior. When measured using constant speed AFM-SMFS in this anchor geometry, the prevalences of the two unbinding pathways were dependent on the force loading rate. The low force pathway was dominant at low loading rates, and as the loading rate increased the high force pathway dominated, precisely resembling a catch bond. We developed kinetic models to describe the response of the DocG:CohE complex to external pulling forces loaded in the catch and slip bond anchor geometries. Monte-Carlo simulations were carried out using the experimental Bell−Evans $\Delta x$ and $k_O$ values as input to simulate the behavior of the complex in constant

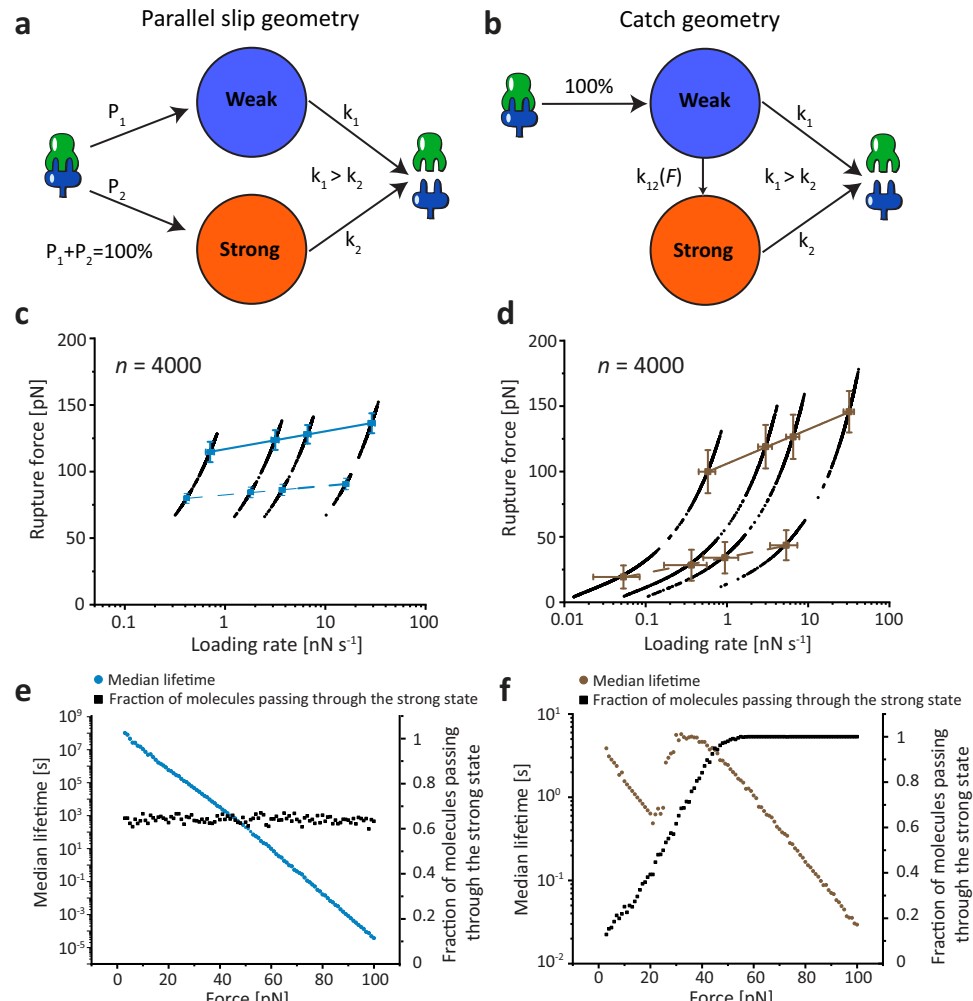

**Fig. 4 | Kinetic model and Monte Calo simulations. a** The complex dissociates in one or two non-interchangeable pathways in the slip bond anchor geometries. The probability of entering each pathway and the energy landscape of each pathway are dependent on the anchor geometry. **b** In the catch bond anchor geometry (pulling from residue 13 of CohE and N-terminus of DocG), the complex enters a weak state upon binding. It either dissociates from the weak state or enters a strong state, which has higher mechanostability and lower off rate than the weak state. The rate of entering the strong state from the weak state is force-dependent and increases with increasing force. Monte Carlo simulated force-loading rate plots of the native

(**c**) and catch (**d**) anchor geometries. The average rupture forces of the strong (solid line) and weak (dashed line) pathways were linearly fitted against the loading rate to extract the $k_O$ and $\Delta x^\ddagger$ values (Supplementary Table S3). The simulation for each anchor geometry was replicated for 4000 times ($n = 4000$) at four different pulling speeds. The error bars represent standard deviation of rupture force and loading rate at a given pulling speed. Monte Carlo simulated median lifetime of the complex at the native (**e**) and the catch (**f**) anchor geometries plotted against force. Black points show the relationship between the probability of unbinding in the strong pathway and the clamping force.

pulling speed measurements. The simulations generated good agreement with experimental data for the rupture force and force-loading rate dependency, demonstrating that the Bell–Evans model is suitable for our system. Furthermore, force clamp Monte Carlo simulations further demonstrated that, in the catch anchor geometry, the lifetime of the complex increased with clamping force in the range of ~20–30 pN, consistent with a catch bond under constant force perturbation. In the native anchor geometry, the complex behaved like a slip bond, where the lifetime decreases exponentially with the clamping force.

Other previously reported DocG:CohE complexes with multiple unbinding pathways typically exhibit dual-binding mode behavior and/or use an adjacent X-module domain to allosterically regulate their mechanostabilities[11,34,50]. In contrast, the *Rc*. DocG:CohE complex lacks an X-module, and populates only one single binding conformation, as demonstrated by single-molecule FRET measurements. The molecular mechanisms underlying the two distinct unbinding pathways of the *Rc*. DocG:CohE complex could be further clarified in future studies involving molecular dynamics simulations and structure determination approaches.

In conclusion, we used a mechanical anisotropy-based protein engineering approach to alter the behavior of a human gut bacterial adhesion protein complex, *Rc*. DocG:CohE, under external pulling forces. By changing the anchor geometry, we could switch the complex between a slip bond and catch bond without making any modifications to the binding interface or influencing the equilibrium binding affinity. This approach can be readily applied to other protein-ligand systems including cell adhesion proteins, antibody-antigen complexes, and alternative scaffolds. Using the mechanical anisotropy-based molecular engineering tool to introduce new mechanical properties may prove beneficial for designing synthetic systems such as biomimetic materials and nanoparticle-based therapies that resist shear flow[61–63].

## Methods
### Structural modeling
The structural model of DocG was built using the SWISS-MODEL server, available at https://swissmodel.expasy.org/interactive. The template was searched with BLAST against the SWISS-MODEL template

library and the selected template structure was PDB 4WKZ (sequence identity 29.11%, sequence similarity 33%, coverage 94%). The structural model was built based on sequence alignment between the target and the template using ProMod3[38]. The structural model of CohE in complex with XMod-DocB was previously reported[11]. The structure model of DocG:CohE complex was built by aligning the structural model of DocG to the XMod-DocB:CohE structure using PyMol.

## Protein expression and purification

The CohE F13AzF and CohE K87AzF mutants, as well as the CohE (F13AzF)-FLN-ELP-ybbr construct, were expressed using amber codon suppression. C321.ΔA.exp bacteria (gift from George Church, Addgene bacterial strain # 49018)[64] were co-transformed with a pQE80l vector carrying the gene of the mutant CohE construct with an amber codon at the desired anchor point, and pEVOL-pAzF plasmid (gift from Peter Schultz, Addgene plasmid # 31186)[59]. The transformed cells were grown in LB medium at 37 °C with ampicillin and chloramphenicol until OD reached ~0.5. Arabinose was added to the medium to a final concentration of 0.02% and the culture was incubated at 37 °C for 1 h, followed by addition of isopropyl β-d-1-thiogalactopyranoside (IPTG) to a final concentration of 1 mM to induce protein expression. The culture was subsequently incubated at 20 °C overnight.

The other proteins were expressed using NiCo21 (DE3) bacterium cells (New England Biolabs, catalog# C2529H). The bacterial competent cells were transformed using pET28a vector carrying the gene of interest and grown in LB medium at 37 °C in the presence of 50 μg mL$^{-1}$ kanamycin until OD reached ~0.6. IPTG was added to the medium to a final concentration of 0.5 mM to induce protein expression. The culture was subsequently incubated at 20 °C overnight.

To purify the proteins of interest, the bacterial cells were pelleted, resuspended in TBS Ca buffer (25 mM Tris, 72 mM NaCl, 1 mM CaCl$_2$, pH 7.2), and lysed using sonication. The cell lysate was centrifuged at 4 °C 18,000 × $g$ for 30 min. The supernatant was loaded to a 5 mL His-trap FF column (Cytiva, Marlborough, MA, United States, catalog# 17525501) and washed using TBS Ca buffer with 20 mM imidazole. The target protein was eluted using TBS Ca buffer with 500 mM imidazole. The eluent was subsequently purified using Superdex 200 Increase 10/300 GL size-exclusion column (Cytiva, catalog# 28990944) in TBS Ca buffer.

## Conjugation of Fgβ with AzF-incorporated CohE

CohE with incorporated AzF (F13AzF or K87AzF) was mixed with 5× molar excess of Fgβ-StrepTag-DBCO peptide (NEEGFF-SARGHRPLDGSWSHPQFEKGSGSC-DBCO, JPT Peptide Technologies GmbH, Berlin, Germany). The reaction mixture was incubated at room temperature with shaking for 1 h, followed by incubation at 4 °C overnight. The reaction mixture was loaded to Superdex 200 Increase 10/300 GL size-exclusion column (Cytiva, catalog# 28990944) and eluted using TBS Ca buffer to remove the excess peptide. The CohE protein purified by SEC was then loaded to a 1 mL StrepTrap HP column (Cytiva, catalog# 28907546) and eluted using TBS Ca buffer supplemented with 2.5 mM desthiobiotin to remove any CohE that was not conjugated with Fgβ.

## Constant pulling speed AFM measurement

Constant pulling speed AFM measurements were performed on a Force Robot AFM (Bruker, Billerica, MA, USA), using JPK SPM Nano-Wizard software version 6.4.21. Biolever mini (Olympus) AFM cantilevers were amino-silanized with (3-aminopropyl)-dimethyl-ethoxysilane (APDMES, ABCR GmbH, Karlsruhe, Germany, catalog# AB146193), incubated in 10 mg/mL sulfosuccinimidyl 4-(N-maleimidomethyl)cyclohexane-1-carboxylate (sulfo-SMCC, Thermo Fisher Scientific, Waltham, MA, USA, catalog# A39268) at room temperature for 30 min, extensively washed with ddH$_2$O, followed by incubation in 200 μM coenzyme A (CoA, Sigma-Aldrich, St. Louis, MO, USA, catalog#

C3144) at room temperature for 2 h. The ybbr-tagged protein (CohE or SdrG) was immobilized on the cantilever by incubating the CoA coated cantilever with a reaction mixture consisting of ~40 μM ybbr-tagged protein, 5 μM Sfp (phosphopantetheinyl transferase) enzyme and 20 mM MgCl$_2$, at room temperature for 2 h.

The cover glasses were silanized and coated with CoA using the same protocol as the cantilevers. CoA-coated cover glasses were incubated with 500 nM ybbr-tagged DocG in the presence of 5 μM Sfp and 20 mM MgCl$_2$ at room temperature for 2 h to immobilize the Doc on the cover glasses.

The cover glasses and AFM cantilever coated with target proteins were extensively washed using TBS Ca buffer and submerged in TBS Ca buffer for AFM measurements. The cantilever spring constants (ranging from 0.02 N m$^{-1}$ to 0.14 N m$^{-1}$) and detector sensitivity were calibrated using the contact-free mode of the Force Robot AFM. In the measurements using freely diffusing CohE, ~100 nM Fgβ-conjugated CohE was added to the measurement buffer. The cantilever was approached to the glass surface, dwelled for 200 ms and retracted at a constant speed ranging from 100 nm s$^{-1}$ to 3200 nm s$^{-1}$ (100 nm s$^{-1}$, 400 nm s$^{-1}$, 800 nm s$^{-1}$ and 3200 nm s$^{-1}$). The x-y position of the sample stage was moved by 100 nm after the retraction, so that a new molecule was probed at the next approach.

## AFM data analysis

The recorded force-extension curves were exported using JPK SPM Data Processing software version 6.4.21, and transformed to contour length using the freely rotating chain (FRC) mode described by Eq. (1)[46]:

$$L = \begin{cases} \frac{3k_BT}{xFa} & \text{for } \frac{Fb}{k_BT} < \frac{b}{l} \\ \frac{x}{1-\left(\frac{4Fl}{k_BT}\right)^{-0.5}} & \text{for } \frac{b}{l} < \frac{Fb}{k_BT} < \frac{l}{b} \\ \frac{x}{1-\left(\frac{2Fb}{k_BT}\right)^{-1}} & \text{for } \frac{Fb}{k_BT} > \frac{l}{b} \end{cases} \quad (1)$$

where $a = b\frac{1+\cos\gamma}{(1-\cos\gamma)\cos\frac{\gamma}{2}}$ is the Kuhn length and $l = b\frac{\cos\frac{\gamma}{2}}{|\ln\cos\gamma|}$ is the persistence length. The segment length $b = 0.11$ nm and bond angle $\gamma = 41°$ were used for the calculation.

The force-extension curves were subsequently filtered based on the stretching of the two ELP linkers on the AFM tip and the glass surface. The final rupture forces and force loading rates were calculated for each of the selected curves. The rupture forces measured at different pulling speeds were plotted in histograms using Origin Pro 2021 software and fitted with one or two Gaussian peaks, depending on the anchor geometry. The area under the high force peak is divided by the sum of the areas under the high force and low force peaks to calculate the prevalence of the strong unbinding pathway. The standard deviation of the prevalence of the strong unbinding pathway was calculated by Gaussian error propagation based on the fitting results of the areas under the high force and low force peaks. The most probable rupture force of each pathway was plotted against the logarithm of loading rate and fitted using a linear model to extract $k_O$ and $\Delta x^{\ddagger}$, as described by the Bell–Evans model (Eq. (2))[18,19]:

$$F^* = \frac{k_BT}{\Delta x^{\ddagger}} \ln\left[\frac{r\Delta x^{\ddagger}}{k_0 k_BT}\right] \quad (2)$$

*where $F^*$ is the most probable rupture force at different loading rates.*

## Fluorescent labeling of CohE and DocG for smFRET

CohE E154C mutant was labeled with Alexa Fluor 647 (Thermo Scientific catalog# A20347). DocG S54C and DocG A77C mutants were labeled with Cy3b (Cytiva catalog# PA63131). The mutants were stored in TBS Ca buffer at a concentration of ~50 μM. The cysteines carried by the proteins were reduced by incubating with 5 mM DTT at room

temperature for 30 min. The proteins were subsequently loaded to a Superdex 200 Increase 10/300 GL size-exclusion column (Cytiva catalog# 28990944) and eluted using PBS buffer (pH 7.4) to remove the DTT. The reduced proteins were incubated with 10x molar excess of maleimide-conjugated fluorophores for 1 h, followed by incubation at 4 °C overnight. The proteins were then supplemented with 10 mM DTT to quench the unreacted maleimide groups and purified using a Superdex 200 Increase 10/300 GL size-exclusion column (Cytiva catalog# 28990944) and eluted in TBS Ca buffer to remove the excess fluorophores.

### Microscale thermophoresis (MST) measurements

Titration samples were prepared by mixing 20 nM DocG A77C mutant protein labeled with Cy3b with a series of CohE wild-type, CohE F13AzF-Fgβ, or CohE K87AzF-Fgβ proteins. 16 samples were prepared for each titration series with the Coh concentration ranging between 0.0763 nM and 2.5 μM. The microscale thermophoresis (MST) traces of each sample were measured using a Nanotemper Monolith NT.115 (NanoTemper Technologies GmbH, Munich, Germany). The temperature-related fluorescence intensity change between the cold range (1 s before measurement started) and the hot range (0.5 s to 1.5 s after the measurement started) was recorded for each titration sample and plotted against the CohE concentration to calculate the affinity of the DocG:CohE complex.

### smFRET measurement chamber passivation

For smFRET solution measurements, Labtek chambers (Lab-Tek II chambered coverglass system, ThermoFisher Scientific, MA, USA) were passivated with 1 mg/ml BSA (PAA laboratories GmbH, Germany). The BSA solution was removed just before the measurement and never before 1 h of incubation. Finally, the chamber was washed out three times with PBS and once with the measurement buffer (50 mM Tris, 100 mM NaCl, 2 mM Trolox/Trolox quinone, 1% glucose, 1 mM CaCl₂, pH 8).

### Confocal smFRET experiments

Confocal solution smFRET experiments were performed on a PIE-based[65] home built confocal microscope based on an Olympus IX-71 inverted microscope. Two pulsed lasers (639 nm, 80 MHz, LDH-D-C-640; 532 nm, 80 MHz, LDH-P- FA-530B; both PicoQuant GmbH) were altered on the nanosecond timescale by a multichannel picosecond diode laser driver (PDL 828 "Sepia II", PicoQuant GmbH, Berlin, Germany) with an oscillator module (SOM 828, PicoQuant GmbH). The lasers were coupled into a single mode fiber (P3-488PM-FC, Thorlabs GmbH, Dachau, Germany) to obtain a Gaussian beam profile. Circular polarized light was obtained by a linear polarizer (LPVISE100-A, Thorlabs GmbH) and a quarter-wave plate (AQWP05M- 600, Thorlabs GmbH, Dachau, Germany). The light was focused by an oil-immersion objective (UPLSAPO100XO, NA 1.40, Olympus Deutschland GmbH) onto the sample. The sample was moved by a piezo stage (P-517.3CD, Physik Instrumente (PI) GmbH & Co. KG, Karlsruhe, Germany) controlled by a E-727.3CDA piezo controller (Physik Instrumente (PI) GmbH & Co. KG, Karlsruhe, Germany). The emission was separated from the excitation beam by a dichroic beam splitter (z532/633, AHF analysentechnik AG, Tübingen, Germany) and focused onto a 50 μm pinhole (Thorlabs GmbH). The emission light was split by a dichroic beam splitter (640DCXR, AHF analysentechnik AG) into a green (Brightline HC582/75, AHF analysentechnik AG; RazorEdge LP 532, Laser 2000 GmbH, Weßling, Germany) and red (Shortpass 750, AHF Analysentechnik AG; RazorEdge LP 647, Laser 2000 GmbH) detection channel. Emission was focused onto avalanche photo diodes (SPCM-AQRH−14-TR, Excelitas Technoligies GmbH & Co. KG, Wiesbaden, Germany) and signals were registered by a time-correlated single photon counting (TCSPC)-unit (HydraHarp400, PicoQuant GmbH, Berlin, Germany). The setup was controlled by a commercial software

package (SymPhoTime64, Picoquant GmbH, Berlin, Germany). Excitation powers of 33 μW and 30 μW were used for donor and acceptor lasers (as measured in front of the entrance of the microscope).

The dye-labeled CohE and DocG samples were first incubated for a few seconds at a concentration of 0.8 μM, at a molar ratio of 1:1. The samples were finally diluted to a concentration of 400 pM in the smFRET buffer (50 mM Tris, 100 mM NaCl, 2 mM Trolox/Trolox quinone, 1% glucose, 1 mM CaCl₂, pH 8).

### smFRET data analysis

Burst selection was performed using a sliding time window burst search, with a time window of 500 μs, a minimum of 4 photons per time window, and a threshold for burst detection of 40 photons[57]. The ALEX-2CDE[66] and |TDX-TAA| filters[67] were applied to sort out photobleaching and blinking events. Furthermore, doubled-labeled molecules were selected by selecting those burst with the stoichiometry parameter between 0.2 and 0.8. The accurate FRET efficiencies[56,68] were calculated using the fluorescence intensities as:

$$E = \frac{I_{DA} - \alpha I_{DD} - \delta I_{AA}}{\gamma I_{DD} + I_{DA} - \alpha I_{DD} - \delta I_{AA}} \tag{3}$$

where $I_{AA}$, $I_{DD}$ and $I_{DA}$ are the background-corrected photon counts in the acceptor channel after acceptor excitation, in the donor channel after donor excitation, and in the acceptor channel after donor excitation. The α and δ correction factors were calculated from donor and acceptor only subpopulations respectively and account for spectral cross talk of the donor and direct excitation of the acceptor dye. The different quantum yields and detection efficiencies of the fluorophores are corrected with the γ correction factor obtained from global fits of $1/S$ vs $E$ plots[56,68]. When proximity ratios, E(PR), are calculated these parameters are set to α = 0, δ = 0, and γ = 1. For the PDA analysis the data was fully corrected.

### Photon distribution analysis (PDA)

Quantitative PDA analysis[58,69,70] was carried out using the free software PAM[71]. For the fit, a Förster radius of $R_0 = 69.12$ Å was used, as calculated using the overlap integrals from donor emission spectra (Cy3b) and acceptor absorption spectra (Alexa Fluor 647). The direct excitation correction factor for the PDA analysis (0.049) was calculated from the extinction coefficients of the acceptor and donor dyes as described in the PAM documentation (https://pam.readthedocs.io/en/latest/pda.html#the-correction-parameters). The time binning was set to 0.8 ms. Fitting parameter errors represent confidence intervals at 95%.

### Monte Carlo simulations

To validate the kinetic models of slip and catch anchor geometries, a Monte Carlo approach based on Kramers theory was used and realized using Python code. To initiate the simulation in the slip anchor geometries (Fig. 4a), the complex was randomly assigned a binding state to be either strong state (64% possibility) or weak state (36% possibility), according to the average prevalence of each pathway from the AFM observation. The corresponding kinetic parameters ($k_O$ and $\Delta x^{\ddagger}$, see Table 1) extracted from AFM-SMFS were used for the simulation. For the catch anchor geometry (Fig. 4b) the complex always starts in weak state, with an extra irreversible pathway that describes the transition from weak to strong state ($k_{12}$ in Fig. 4b). The corresponding kinetic parameters for $k_{12}$ were obtained based on the dataset containing four pulling speeds and the corresponding pathway prevalence (η), using a nonlinear least-square fitting approach as previously described[60]. In short, simulations were based on adjustable parameters (in this case, Bell−Evans $k_O$ and $\Delta x^{\ddagger}$ for $k_{12}$) were conducted to yield theoretical $\eta^*$ and the $\eta$ residual could be calculated comparing $\eta^*$ with experimentally observed η. This process was repeated using iteratively updated energy landscape parameters until the tolerance on η

residuals was reached. For the catch anchor geometries, an energy profile for the weak to strong pathway transition of $\Delta x^{\ddagger} = 0.83$ nm, $k_O = 0.019$ s$^{-1}$ was obtained.

To simulate the constant speed protocol on AFM, a series of force values $F(t_i)$ was generated on an evenly distributed molecular extension axis $X(t_i)$ using a worm-like chain (WLC) model[72], followed by a bending correction that converts the molecular extension to the AFM head height $H(t_i)$ where the constant pulling speed is applied using Eq. (4), where k represents the spring constant of AFM cantilever. Then the time series $t_i$ could be generated based on the pulling speed $V$ following Eq. (5).

$$H(t_i) = X(t_i) + \frac{F(t_i)}{k} \quad (4)$$

$$t_{i+1} = t_i + \frac{H(t_{i+1}) - H(t_i)}{V} \quad (5)$$

Within each time interval, the probability of the complex rupture and the weak to strong state transition in catch anchor geometries could be calculated from the given energy profile using the following equation:

$$P(F) = 1 - e^{-k_{off}(F)\Delta t} \quad (6)$$

where the force dependent off-rate $k_{off}(F)$ can be drawn from Eq. (7) following the Bell–Evans model:

$$k_{off}(F) = k_0 e^{\beta F \Delta x^{\ddagger}} \quad (7)$$

where $\beta^{-1} = k_B T$. The obtained probability is compared to an independent random number between zero and unity, to check if the stochastic event of complex rupture, or the weak to strong state transition happens. For each pulling speed, 1000 curves were generated and a histogram was drawn for the complex rupture force. For force clamp conditions, a constant force was applied and 1000 curves were generated to calculate the median lifetime of the complex under each force setpoint.

### Reporting summary
Further information on research design is available in the Nature Portfolio Reporting Summary linked to this article.

## Data availability
Experimental data from single-molecule AFM and single-molecule FRET experiments in the main text and supplemental information, as well as Monte Carlo simulation data is publicly available in Zenodo [https://zenodo.org/records/10784782]. Source data are provided with this paper.

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

## Acknowledgements

This work was supported by the University of Basel, ETH Zurich, an ERC Starting Grant (MMA-715207), the NCCR in Molecular Systems Engineering, and the Swiss National Science Foundation (Project 200021_175478). A.M.V. thanks the Munich Multiscale Biofabrication Network for the financial support. The authors thank Peter Schultz for providing the pEVOL-pAzF plasmid (Addgene plasmid # 31186) and George Church for providing the C321.ΔA.exp bacterial cells (Addgene bacterial strain # 49018). The authors thank the Biophysics Facility of Biozentrum, University of Basel, for the help with affinity measurements using MST.

## Author contributions

Z.L. and M.A.N. conceived the study. Z.L. and B.Y. prepared biological samples. Z.L. and H.L. performed AFM-SMFS experiments and analyzed the data. A.M.V. performed single-molecule FRET experiments and analyzed the data. H. L. performed Monte Carlo simulations. P.T. and M.A.N. supervised the project. Z.L. and M.A.N. drafted and edited the manuscript with input from all authors. All authors reviewed the manuscript.

## Competing interests

The authors declare no competing interests.
