## [Peer Review File · Nature Communications]

Engineering an artificial catch bond using mechanical anisotropyEditorial Note: This manuscript has been previously reviewed at another journal that is not operating a transparent peer review scheme. This document only contains reviewer comments and rebuttal letters for versions considered at *Nature Communications*.

REVIEWERS' COMMENTS

Reviewer #2 (Remarks to the Author):

The authors have made a substantial effort to address the comments by the reviewers. They have introduced numerous improvements throughout the manuscript. In particular, the plot of the simulation results together with the experimental data suggests that the reported interpretation as anchor-point-dependent catch-slip bond is solid. The change that the authors made to Figure 4 illustrates the mechanism better. Overall, the work represents a beautiful contribution to the field because it demonstrates experimentally how a catch bond can be engineered starting from a slip bond.

Reviewer #4 (Remarks to the Author):

In this manuscript, the authors use AFM-SMFS to show that the Dockerin G:Cohesion E adhesion complex, a bacterial protein complex that behaves as a slip bond in its native state, can be artificially engineered to exhibit catch bonding behavior depending on the specific residues through which the complex is anchored and pulled.

Dockerin G (DocG) and Cohesion E (CohE) were expressed as fusion proteins with a terminal ybbr tag, an ELP spacer, and an FLN domain, mimicking native bacterial geometry. DocG and CohE were immobilized onto a glass substrate and AFM tip, respectively, through a ybbr-CoA interaction. The complex was associated and dissociated through AFM cantilever actuation at four pulling speeds and plotted in histograms, revealing bimodal distributions in all cases which suggest that the complex exhibits two different unbinding pathways.

To confirm the existence of two different unbinding pathways, which has been shown in several other Doc:Coh complexes, a single-molecule FRET experiment was employed to measure relative distances between specifically labelled residues. Based on structural modeling, CohE was labelled with AF647 at E154 on its C-terminus end while DocG was labelled with Cy3b either at S54 on helix 2 or A77 on its C-terminus end. To covalently label these residues, they were mutated to cysteines and maleimide-fluorophores were used to conduct bioconjugation reactions. CohE-E154C-AF647 was mixed with both of the DocG Cy3b derivatives and PDA was used to fit the distance between FRET pair AF647 and Cy3b. In both cases, unimodal photon distributions were observed, indicating that the binding of the DocG:CohE complex exhibits one binding mode (via this FRET experiment) and two competing unbinding modes (via earlier AFM experiment). The distance between CohE-E154 and DocG-A77 is shorter than CohE and the other DocG derivative, indicating that "conformation A" is the dominant binding conformation of this complex. The authors propose two kinetic models to describe the complex's catch vs. slip geometries. When the complex binds in the catch geometry, it enters a weak binding state and, when pulled, either immediately unbinds or transitions to a strong state from which it can subsequently unbind. The force at which the complex is pulled dictates whether it will unbind at a weak or strong external force. Monte

Carlo simulations of the process validate experimental results.

The manuscript is succinct, tells a clear story, and the basic findings are novel. The experimental results gave rise to the hypothesis that this complex might be a valid candidate for engineering catch bond behavior based on a non-natural anchoring geometry, based on the observed its ability to switch between high-force and low-force dissociation at different pulling rates. Five geometries were tested, where either the CohE or DocG would be adhered to the substrate or AFM tip through its native geometry and the binding partner would be adhered through a non-native position. Non-native anchor geometries were produced by mutating residues at several selected positions by introducing non-native azide bearing residue and click conjugating the resulting protein to streptag-Fg β -DBCO. The resulting conjugates were purified by affinity chromatography through the streptag. Fg β -CohE conjugates were immobilized to the AFM tip through binding to a SdrG anchor motif. Following AFM pulling experiments, it was found that anchoring CohE at F13 and DocG at its N-terminus resulted in catch bonding behavior.

Importantly, this work demonstrates how minor sequence modifications to a protein complex can produce a molecular architecture that mechanically behaves considerably different than its native state. The authors have taken previous reviewer comments into consideration and addressed all major concerns. In its current state, the manuscript just needs a bit of minor adjusting to improve its readability. The manuscript should be published after the following comments are addressed:

1. There are some errors in the Figure 2 caption and associated text. The figure caption does not contain any reference to what has been labelled as “C” in the image. I assume what you have labelled as “B: smFRET measurements...” should be labelled “C”. Additionally, please make sure the associated text description in the “Single-molecule FRET shows a single binding conformation” section reflects this.
2. Some of the previous reviewers have suggested that the names of specific complexes and the residue mutations be edited for clarity. The authors have addressed these comments, but I still believe that the manuscript generally lacks significant clarity around this point. Table 1 is helpful, but still takes some mental gymnastics to parse. Consider separating the “Anchor geometry” column into two columns, one that describes DocG’s geometry and another that describes CohE’s geometry. Then maybe give each of the configurations a number or letter abbreviation (like one would see in a synthetic chemistry manuscript where the author refers to a chemical structure as a number or letter they have previously defined in a scheme).
3. Figure 3D is graphically very hard to digest. Overlaying average points with error bars on each cluster of raw data points AND fit lines is very hard to decipher. At minimum, consider selecting colors that can be better discerned from each other. Otherwise, this figure does little to add to the overall story of the paper. Additionally, consider changing the X-axis labels of Figure 3E to the abbreviations suggested in Comment 2. Figure S6 could benefit from a similar restructuring – please find some different colors to replace the fit lines that make them distinct from the underlying raw data points.

Reviewer #4

In this manuscript, the authors use AFM-SMFS to show that the Dockerin G:Cohesion E adhesion complex, a bacterial protein complex that behaves as a slip bond in its native state, can be artificially engineered to exhibit catch bonding behavior depending on the specific residues through which the complex is anchored and pulled.

Dockerin G (DocG) and Cohesion E (CohE) were expressed as fusion proteins with a terminal ybbr tag, an ELP spacer, and an FLN domain, mimicking native bacterial geometry. DocG and CohE were immobilized onto a glass substrate and AFM tip, respectively, through a ybbr-CoA interaction. The complex was associated and dissociated through AFM cantilever actuation at four pulling speeds and plotted in histograms, revealing bimodal distributions in all cases which suggest that the complex exhibits two different unbinding pathways.

To confirm the existence of two different unbinding pathways, which has been shown in several other Doc:Coh complexes, a single-molecule FRET experiment was employed to measure relative distances between specifically labelled residues. Based on structural modeling, CohE was labelled with AF647 at E154 on its C-terminus end while DocG was labelled with Cy3b either at S54 on helix 2 or A77 on its C-terminus end. To covalently label these residues, they were mutated to cysteines and maleimide-fluorophores were used to conduct bioconjugation reactions. CohE-E154C-AF647 was mixed with both of the DocG Cy3b derivatives and PDA was used to fit the distance between FRET pair AF647 and Cy3b. In both cases, unimodal photon distributions were observed, indicating that the binding of the DocG:CohE complex exhibits one binding mode (via this FRET experiment) and two competing unbinding modes (via earlier AFM experiment). The distance between CohE-E154 and DocG-A77 is shorter than CohE and the other DocG derivative, indicating that "conformation A" is the dominant binding conformation of this complex. The authors propose two kinetic models to describe the complex's catch vs. slip geometries. When the complex binds in the catch geometry, it enters a weak binding state and, when pulled, either immediately unbinds or transitions to a strong state from which it can subsequently unbind. The force at which the complex is pulled dictates whether it will unbind at a weak or strong external force. Monte Carlo simulations of the process validate experimental results.

The manuscript is succinct, tells a clear story, and the basic findings are novel. The experimental results gave rise to the hypothesis that this complex might be a valid candidate for engineering catch bond behavior based on a non-natural anchoring geometry, based on the observed its ability to switch between high-force and low-force dissociation at different pulling rates. Five geometries were tested, where either the CohE or DocG would be adhered to the substrate or AFM tip through its native geometry and the binding partner would be adhered through a non-native position. Non-native anchor geometries were produced by mutating residues at several selected positions by introducing non-native azide bearing residue and click conjugating the resulting protein to streptag-Fgβ-DBCO. The resulting conjugates were purified by affinity chromatography through the streptag-Fgβ-CohE conjugates were immobilized to the AFM tip through binding to a SdrG anchor motif. Following AFM pulling experiments, it was found that anchoring CohE at F13 and DocG at its N-terminus resulted in catch bonding behavior.

Importantly, this work demonstrates how minor sequence modifications to a protein complex can produce a molecular architecture that mechanically behaves considerably different than its native state. The authors have taken previous reviewer comments into consideration and addressed all

major concerns. In its current state, the manuscript just needs a bit of minor adjusting to improve its readability. The manuscript should be published after the following comments are addressed:

We thank the reviewer for carefully reading our manuscript and the encouraging and constructive comments.

1. There are some errors in the Figure 2 caption and associated text. The figure caption does not contain any reference to what has been labelled as “C” in the image. I assume what you have labelled as “B: smFRET measurements...” should be labelled “C”. Additionally, please make sure the associated text description in the “Single-molecule FRET shows a single binding conformation” section reflects this.

We have corrected the figure caption.

2. Some of the previous reviewers have suggested that the names of specific complexes and the residue mutations be edited for clarity. The authors have addressed these comments, but I still believe that the manuscript generally lacks significant clarity around this point. Table 1 is helpful, but still takes some mental gymnastics to parse. Consider separating the “Anchor geometry” column into two columns, one that describes DocG’s geometry and another that describes CohE’s geometry. Then maybe give each of the configurations a number or letter abbreviation (like one would see in a synthetic chemistry manuscript where the author refers to a chemical structure as a number or letter they have previously defined in a scheme).

We have revised Table 1 to describe the DocG and CohE geometries separately. We also used letter abbreviations to refer to the non-native anchor geometries.

3. Figure 3D is graphically very hard to digest. Overlaying average points with error bars on each cluster of raw data points AND fit lines is very hard to decipher. At minimum, consider selecting colors that can be better discerned from each other. Otherwise, this figure does little to add to the overall story of the paper. Additionally, consider changing the X-axis labels of Figure 3E to the abbreviations suggested in Comment 2. Figure S6 could benefit from a similar restructuring – please find some different colors to replace the fit lines that make them distinct from the underlying raw data points.

We have removed the raw data points from Figure 3D, and changed the x-axis labels of Figure 3E to the letter abbreviations.